# Exploring Mammalian Genome within Phase-Separated Nuclear Bodies: Experimental Methods and Implications for Gene Expression

**DOI:** 10.3390/genes10121049

**Published:** 2019-12-17

**Authors:** Annick Lesne, Marie-Odile Baudement, Cosette Rebouissou, Thierry Forné

**Affiliations:** 1IGMM, Univ. Montpellier, CNRS, F-34293 Montpellier, France; marie-odile.baudement@nmbu.no (M.-O.B.); cosette.rebouissou@igmm.cnrs.fr (C.R.); 2Sorbonne Université, CNRS, Laboratoire de Physique Théorique de la Matière Condensée, LPTMC, F-75252 Paris, France; 3Centre for Integrative Genetics (CIGENE), Faculty of Biosciences, Norwegian University of Life Sciences, 1430 Ås, Norway

**Keywords:** phase separation, nuclear bodies, self-assembly, genome organization, gene expression

## Abstract

The importance of genome organization at the supranucleosomal scale in the control of gene expression is increasingly recognized today. In mammals, Topologically Associating Domains (TADs) and the active/inactive chromosomal compartments are two of the main nuclear structures that contribute to this organization level. However, recent works reviewed here indicate that, at specific loci, chromatin interactions with nuclear bodies could also be crucial to regulate genome functions, in particular transcription. They moreover suggest that these nuclear bodies are membrane-less organelles dynamically self-assembled and disassembled through mechanisms of phase separation. We have recently developed a novel genome-wide experimental method, High-salt Recovered Sequences sequencing (HRS-seq), which allows the identification of chromatin regions associated with large ribonucleoprotein (RNP) complexes and nuclear bodies. We argue that the physical nature of such RNP complexes and nuclear bodies appears to be central in their ability to promote efficient interactions between distant genomic regions. The development of novel experimental approaches, including our HRS-seq method, is opening new avenues to understand how self-assembly of phase-separated nuclear bodies possibly contributes to mammalian genome organization and gene expression.

## 1. Introduction

Several physical properties of nuclear organization are critical for regulating mammalian gene expression. In interphase, the genome is highly compacted to fit into the limited space of the cell nucleus while, at the same time, it remains fully accessible to multiple interactions involving *cis*- and *trans*-acting genomic elements and RNA/protein factors. Such a paradoxical achievement of a compact but dynamic genome is solved not only by packaging the genome into the chromatin nucleofilament, but also through a complex compartmentalization of the nucleus that contributes to the functional genome organization at the supranucleosomal scale (i.e., encompassing few tenths of kb to few Mb of DNA). The functional role of 3D genome organization has thus become an important component in the study of mammalian gene expression [1].

Another paradigm has been recently re-examined and developed: biomolecular condensates, grounded in the classical physical notion of phase separation [2]. While the use of this concept in a biological context dates back the old notion of coacervate, its relevance has been recently renewed by technological advances allowing in vivo observations and mechanistic investigations [3]. 

Phase separation describes the spontaneous formation of a two-phase system. From a physical point of view, it covers not only the demixing of oil and water, but also the spatial segregation that can arise in aqueous solutions, when the attraction between the solute molecules is energetically favored compared to the interaction between these molecules and the aqueous solvent. The balance between interaction energies and thermal motion or the ensuing diffusion, described by the free energy of the system, can lead in appropriate conditions to the spatial segregation of two phases of different concentrations [4]. This phenomenon is known as liquid–liquid phase separation (LLPS). Indeed, self-separated droplets display several features of a liquid phase: they are dense (as opposed to gases), display no rigid order (as opposed to crystals or liquid crystals), and their molecules remain mobile (as opposed to solids and gels), with permanent exchange between the two phases. These droplets display fluid-like behaviour, in particular the fusion of adjacent droplets into larger ones and a shape determined by surface tension. However, their composition, particularly under biological constraints, make them far more complicated than a mere liquid. Experimental strategies are thus developed to assess the presence and specificity of phase separation inside the cell [5]. Noticeably, “condensate” is the term used for molecular assemblies that form through phase separation while the more general term “hub”covers molecules that cluster together through yet unknown mechanisms.

Phase separation has been first recognized in the cytoplasm, as a mechanism of formation of stress granules and P-bodies [4]. It has been more recently invoked in the nucleus, for instance for the formation of membrane-less organelles also known as nuclear bodies. Much work is now devoted to identifying the hallmarks of in vivo phase separation and devising suitable protocols to study it [6]. In this review, we will first examine the proposal that nuclear compartments are phase-separated and could influence transcriptional regulation through their association with specific genomic sequences [7,8]. We will then present a novel experimental approach, HRS-seq, to test this working hypothesis. 

## 2. Compartmentalization of Chromatin Interactions

In the past decade, the advent of sophisticated imaging techniques and molecular biology approaches based on proximity ligation assays (3C/Hi-C) has revealed that beyond the compaction achieved by packaging the DNA molecule at the nucleosomal level, chromatin is also organized within the three-dimensional (3D) space of the nucleus [9,10]. This 3D chromatin folding displays nested features, the most acknowledged being chromatin loops and topologically associating domains (TADs) where preferential *cis*-long-range contacts are observed [11]. A higher-order organization level also exists that partly covers the classic distinction between euchromatin and heterochromatin: the active (A) and inactive (B) chromosomal compartments [12]. While cohesin and CTCF proteins are required for TAD organization, these factors are dispensable for the maintenance of chromosomal compartments, which rely on different organization principle [13,14,15]. Furthermore, while TADs are essential for cell-specific genome organization and function [1], they appear to be quite stable between cell types, and even between organisms along evolution [16,17]. In striking contrast, chromatin loops and chromosomal compartments appear to vary during cell differentiation [18] and therefore they presumably play a central role for establishing specific gene expression profiles that determine cell identities. Several recent works started to decipher some crucial aspects of compartment regulation during mammalian spermatogenesis [19,20,21,22], in oocyte or early embryonic development [23,24,25], during cell differentiation [18,26] or reprograming [27] (for a recent review see [28]). However, to fully understand how 3D genome organization controls mammalian gene expression, it is critical to focus not only on long-range *cis*-interactions occurring at specific loci within TADs but also on *trans*-associations occurring between TADs within chromosomal compartments. 

## 3. Nuclear Body Assembly by Phase Separation

Nuclear bodies are large membrane-less ribonucleoprotein (RNP) complexes known to be involved in several nuclear functions. For example, the synthesis of ribosomal RNAs (rRNAs) takes place in the nucleolus, the maturation of small nuclear RNAs (snRNAs) occurs in the Cajal bodies, and the histone messenger RNAs (mRNAs) are transcribed and matured in the histone-locus bodies (HLBs) (Table 1). One important aspect of functional nuclear compartmentalization is thus related to nuclear bodies. Some of them, like the HLBs, are known to gather loci that are dispersed in TADs located on distinct chromosomes, thus favouring coordinated gene transcription and efficient pre-mRNA maturation [29]. Similarly, the Cajal bodies have also been shown to contain inter-TAD interactions [30]. Transcription factories and active chromatin hubs are also large RNP complexes that have been proposed to coordinate gene expression by maintaining specific genes into a restricted 3D space of the nucleus [31]. Large RNP complexes, including some nuclear bodies, thus appear important for supranucleosomal genome organization in mammals. Indeed, their involvement in regulating transcription of specific genes suggests that they might be critical for the establishment and the maintenance of the active chromosomal compartment. However, the demonstration of such a role has so far been impeded by the lack of a genome-wide method that would allow unbiased profiling of genomic sequences associated with nuclear bodies. In our view, this is due to a continued lack of understanding of the physical nature of nuclear bodies in vivo. 

It has been thought for a long time [41] that nuclear bodies are self-organized around nucleation sites, e.g., the Nucleolar Organizing Regions -NORs- for the nucleolus or the histone H3-H4 promoter region for Drosophila HLBs [42,43,44]. As a precedent, several cytoplasmic components, like the *Caenorhabditis elegans* P-granules [45] and centrosomes [46], have been discovered to behave in vivo like self-organized liquid-like droplets. More recently, based on in vitro experiments, other cytoplasmic structures, like the glycolytic bodies [47] or the RNA granules [48], have been proposed to form by phase separation processes. However, experimental evidence supporting self-organization or self-assembly remained very scarce for nuclear bodies (for reviews see [4,49]). A step forward has been the proposal, based on in vitro reconstitution experiments, that the phase separation of liquid-like RNP phases could control nucleolus size and assembly [50,51], as well as account for their sub-compartmentalized organization [52]. The demonstration that the Intrinsically Disordered Region (IDR) of Ddx4 protein (a critical component of the mammalian analogue to P-granules) can form phase-separated organelles, both in live cells and in vitro [53], led to the more precise hypothesis that phase separation of IDR-containing proteins could be a general mechanism for forming and regulating membrane-less organelles. These pioneering findings paved the way to a number of studies aimed at deciphering whether phase separation is involved in the organization of other nuclear compartments or bodies. 

One can distinguish two phase separation processes: liquid–liquid phase separation (LLPS) and polymer–polymer phase separation (PPPS). While LLPS occurs through demixing of two liquid/liquid-like phases, PPPS involves bridging factors binding onto a polymer, e.g., the chromatin fiber, leading to a polymer collapse (i.e., a change in the polymer shape accompanied with an increase of its local density) [54]. Beyond the intrinsic nature of the interacting molecules responsible for phase separation (bridging factors for PPPS vs. weak multivalent binders for LLPS), the main differences between these two phase-separation processes lie in the role of the underlying polymer, if any. In PPPS, the polymer is required not only to nucleate phase separation but also to maintain it [55]. On the contrary, while an underlying polymer could help nucleation of LLPS, it is dispensable to maintain phase separation once a given saturating concentration of self-associating multivalent binder molecules has been reached [54]. 

Noticeably, phase separation was proposed to be involved in constitutive heterochromatin domain formation, based on the observation that a major component of the heterochromatin, the heterochromatin 1 α (HP1α) protein, can form liquid droplets both in vitro and in vivo [56,57]. HP1 self-oligomerization driven by phosphorylation is sufficient to induce HP1 phase separation in vitro [56]. However, since HP1α compartments can incorporate chromatin [56], the formation of heterochromatin domains in vivo, could actually be more complex [58] and rely not only on LLPS and weak multivalent chromatin binders [56,57], but also on PPPS, where a bridging factors, like the HP1 proteins themselves [59], could also induces a partial collapse of the chromatin [54,58].

## 4. Phase-Separation Models for Transcription Control

Following these discoveries, Phillip Sharp and colleagues proposed a phase-separation model for transcription control, in which a transcriptional multi-molecular assembly (i.e., a transcriptional condensate) would form by phase separation at a given locus following the formation of large RNP complexes induced by the binding of transcription factors at both enhancers and gene promoters [60]. This model was recently reinforced by studies showing that: (i) transcriptional coactivators, like BRD4 and the Mediator complex at active super-enhancers, together with the RNA polymerase II at promoters, can form transcriptional condensates in vitro [61,62], and (ii) domains driving gene activation in vivo are also required for phase separation in vitro [63]. Such transcriptional hubs, however, are relatively small compared to nuclear bodies. Therefore, it is not yet clear if their formation in vivo truly relies on phase separation and, if so, whether it is based on the demixing of two liquid-like phases similar to the LLPS observed for larger nuclear compartments like the nucleolus, or whether it reflects a hybrid situation also involving a polymer collapse process and PPPS as suspected in the case of heterochromatin domains. In all instances, we should remain careful before considering any transcriptional hub as a condensate formed by phase separation. Indeed, on the one hand RNA polymerase II was shown to form clusters or hubs at active genes through electrostatic interactions between its carboxy-terminal domain (CTD), a prominent IDR, and transcriptional coactivators, suggesting that compartmentalization may occur here through a LLPS process [7]. On the other hand, the transient unspecific binding of RNA polymerase II to the largely nucleosome free genome of the Herpes Simplex Virus type 1 (HSV1) leads to a DNA-mediated nuclear compartmentalization through a mechanism that is clearly distinct from LLPS [55]. Moreover, given the relatively small size of these transcriptional hubs, the physical properties that usually characterize the liquid state of the matter (like surface tension) may well make no real physical and biological sense [58,64]. That is precisely why the terms “hub” and “liquid-like phase separation” are often preferred to “condensate” and LLPS respectively [64]. The difference between a liquid and other states of the matter (like crystal, amorphous solid, liquid crystal or gel) lies in the mobility of the molecules, their ordered or disordered arrangement and the response to a stress (elastic vs. viscous). A whole range of intermediary behaviours are possible (e.g., the viscoelastic response of a gel). At the molecular scale, the liquid state is best characterized by the mobility of the molecules which is essentially depending on diffusion. Experimentally, FRAP (fluorescence recovery after photobleaching) experiments are used to quantitatively assess this mobility [50,57,63]. However, several caveats have been raised [5,6], the main one being that there are many physical models that can be fitted to the same fluorescence recovery curves [64]. Indeed, the rate of fluorescence recovery is not always due to freely diffusing molecules in solution, but could also depend on the local binding to others molecules. One critical point is thus to find experimental controls that could demonstrate that, independently of the models, the recovery rate is truly dominated by diffusion rather than binding. It has been proposed that this could be achieved, for example, by showing a dependence of the recovery rate on the size of the bleach spot [65].

In parallel, another work indicated that various IDR-containing proteins form molecular hubs that could selectively associate in vivo with some chromatin regions by physically retaining targeted genomic loci while excluding non-targeted regions [66]. This chromatin filter model suggests that such molecular hubs could bring distal genomic loci together. However, these experiments use a novel CRISPR-Cas9-based technology (CasDrop) to artificially target chimeric IDR-containing proteins to chosen genomic sequences. It remains to be seen whether endogenous IDR-containing proteins act in a similar way on their natural targets. Additional work has shown the potential involvement of RNA-binding proteins [67].

In Figure 1, we provide an integrated model presenting the current working hypothesis, where we combine the concepts proposed in [60,63,66] for phase-separated transcriptional condensates involving long-range *cis*-interactions and extend these concepts to the probable involvement of nuclear bodies favoring inter-TADs *trans*-associations of co-regulated genes, like those observed for HLBs and Cajal bodies [30].

Our present working hypothesis, as synthesized in the integrated model (Figure 1), has two logical consequences: First, studying the physical principles and factors underlying the assembly of phase-separated nuclear bodies should differentiate at least two main classes of genes, those that are contacting phase-separated transcriptional condensates and those that are not, with as many sub-classes as types of condensates that interact with chromatin. Second, there should be at least two classes of membrane-less nuclear compartments, those that are depending (in vivo) on polymer–polymer phase separation (PPPS) and those that are depending on liquid–liquid or liquid-like phase separation (LLPS).

## 5. HRS-seq: A Novel Method to Explore Nuclear Bodies-Associated Sequences

Further exploration of the role of nuclear bodies in genome organization requires, as previously mentioned, an unbiased genome-wide sequencing of nuclear bodies-associated sequences. So far, these sequences have been difficult to analyse because no efficient and reliable method was available to purify nuclear bodies, presumably due to their membrane-less phase-separated nature.

It is known that performing high-salt treatments of transcriptionally active nuclei makes large RNP complexes, including nuclear bodies, insoluble [68]. More recently, we have shown that a 2M NaCl treatment traps the genomic DNA associated with these RNP complexes into the insoluble material which is easily purified on a filtration unit [69]. The trapped DNA fragments, that we named the “High-salt Recovered Sequences” (HRS), can then be separated from the rest of the genome by performing a simple restriction digestion and washing out the soluble material (Figure 2). The HRS thus remain on the filter unlike the rest of the genomic DNA. High-throughput sequencing of the HRS (HRS-seq) is then performed to obtain a global profiling of sequences associated with high-salt insoluble large RNP complexes, including nuclear bodies [70].

Most existing methods such as FAIRE-seq [71], ATAC-seq [72] or MNase-seq [73,74,75] aim at investigating accessibility of the chromatin nucleofilament at the nucleosomal scale. So far, only few approaches, like the HRS-seq, have been developed to investigate higher-order chromatin architecture at the supranucleosomal scale. Those include DamID mapping [76], 3C-derived methods like the Hi-C [12], MAR-seq [77] and TSA-seq [78]. Unlike all of these methods, the HRS-seq is avoiding delicate chemical crosslinking procedures or the use of specific antibodies that may restrict retention of some genomic sequences. Furthermore, it generally displays a better genomic resolution (few kb vs. few hundred kb) and is much straightforward and cheaper than existing approaches. However, in its present form, the HRS-seq method has several important limitations, the first of which is the fact that many large RNP complexes are extracted jointly in the insoluble material. A second limitation is that, contrary to 3C-derived approaches, it does not provide any indication on the physical proximity of the recovered sequences in vivo. Therefore, there is a clear need for improvements that would allow to identify sequences present simultaneously within specific subnuclear compartments. While assessing physical proximity will require to adapt a proximity-ligation assay to the HRS approach, the first limitation can already be addressed indirectly without modifying the existing HRS-seq protocol. Indeed, the inactivation of specific nuclear bodies by CRISPR/Cas9 technologies targeting critical components in relevant cellular models, combined with the present HRS-seq approach comparing wild-type and mutated cells, should soon allow extensive genomic profiling of sequences associated with specific nuclear bodies. This should lead to a much deeper understanding of how nuclear body-associated sequences and linked gene expression are dynamically affected during embryonic development and cellular differentiation, as well as in pathological situations where nuclear body formation is altered. For instance, in Spinal Muscular Atrophy (SMA), mutations of the *survival of motor neuron 1* (*SMN1*) gene affect Cajal bodies formation and lead to motor neuron death [79]. HRS-seq experiments on heathy or SMA-patient motor neurons should thus provide new insights on altered genomic organization and gene expression in the context of defective Cajal bodies.

The two logical consequences presented in the previous section could thus be tested in vivo using the HRS-seq method or quantitative PCR analyses of HRS assays (HRS-qPCR) in appropriate cellular models. Indeed, our recent work in mouse embryonic stem cells showed that HRS include sequences associated with nuclear bodies (like the Cajal bodies, the HLBs, the speckles and paraspeckles). Moreover, transcriptional hubs formed around super-enhancers are also retained in our assay [69]. In full agreement with the first consequence mentioned above, we found that two classes of genes can be defined according to the criterion of their association (or lack thereof) with large high-salt insoluble RNP complexes [69]. Our work showed that HRS-located genes are highly expressed and associated with the active chromosomal compartment and active super-enhancers in a cell-type specific manner, while genes that do not lie in HRS are moderately or weakly expressed.

Testing the second consequence will require experimental differentiation of PPPS from LLPS. As explained above, these two modalities of phase separation differ by the nature of the interacting molecules and the role played by the DNA/chromatin nucleofilament. Therefore, LLPS and liquid-like phase separation should be sensitive to compounds that disturb weak hydrophobic interactions, like moderate 1,6 hexanediol treatments [80], unlike PPPS that relies on stronger interactions. So far, sensitivity to 1,6 hexanediol has provided the best experimental evidence in favor of the involvement of liquid-like phase-separation processes in the assembly of transcriptional condensates in vivo [61,62], as well as for other classical nuclear bodies like the paraspeckles [81]. Therefore, combining 1,6 hexanediol treatments with HRS-seq could identify the genomic content of phase-separated condensates formed by LLPS driven by hydrophobic interactions. In contrast, molecular condensates formed by PPPS or those that would rely on a phase separation purely driven by electrostatic interactions (i.e., interactions between charged molecules, that are not disrupted by 1,6 hexanediol) are expected to be unaffected by 1,6 hexanediol treatment.

## 6. Discussion

The assembly of membrane-less compartments by phase separation appears to be a powerful mechanism for nuclear compartmentalization that could drive inter-TADs interactions between distant specific genomic loci. Such a compartmentalization could be essential to coordinate complex genomic functions, in particular transcription. At the molecular scale, thermal motion involved in phase-separation processes implies a continuous exchange of molecules between the dense and the dilute phases. Phase separation depends on the local concentration within the nucleus (or a region of the nucleus) of critical components, like IDR-containing proteins, and can thus be controlled by regulating their availability. This could be achieved by simple post-translational modifications that affect the protein’s ability to establish multivalent interactions, like phosphorylation [82]. Supporting this, PRKACB (catalytic subunit of PKA cAMP-dependent protein kinase) and HIP (Homeodomain-interacting protein) kinases are required for the in vivo assembly of the Cajal and PML bodies, respectively [83]. However, little is known about nuclear body homeostasis, which certainly constitutes a promising topic for future investigations.

Liquid–liquid phase separation is not a feature involving an isolated molecular species but is depending on the properties of both this species and the solvent. In the case of nuclear bodies, the solvent corresponds to the complex nuclear environment in which the molecular species of interest is considered. A modification of the contents of this environment would thus affect phase separation. Obviously, a structural or chemical modification of the phase separation-prone molecular species would also affect spatial structuration. Various means of tuning the physical process of phase separation are thus possible within a living cell. While in vitro experiments usually monitor physical parameters controlling phase separation (like temperature or pH), in vivo a specific adaptation of the relative strength of the molecular interactions, through some post-translational modification of the phase separation-prone protein, would offer a more precise control of the process.

The current thermodynamic description of phase separation processes is only valid on a large scale (i.e., involving large enough number of molecules). The direct effects of the intrinsic stochasticity prevailing at molecular scales are random binding/unbinding of interacting molecules, diffusion and ensuing concentration fluctuations. They are included only in an average way in the large-scale thermodynamic description. In case of small systems with a finite number of molecules, (e.g., a region of the nucleus), discrepancies may arise, among which a modification of the stability regions in the parameter space, loss of correlations in cooperative assembly, or various diffusion-limited behaviours. Thus, the robustness with respect to molecular noise of a thermodynamically predicted phase separation needs to be investigated. In the spirit of studies quantifying the stochasticity of transcription [84], the analysis of imaging data or measurements obtained from a large number of single cells observed in the same conditions would assess the variability of the phase separation phenomenon. On the theoretical side, the thermodynamic approach could be supplemented with stochastic dynamical equations including a noise term [85] and the simulation of their solutions [60,86].

Finally, to date, investigations have relied on the description of phase separation in the framework of thermodynamic equilibrium. Nevertheless, active processes are possibly at work in vivo. An example is the observation of droplet fission [87] that is not accounted for in the current thermodynamic models of phase separation. Investigating active features of intracellular dynamic organization thus opens a fascinating research field not only for biologists but also for theoretical physicists. Phase separation is actually a special instance of the more general concept of self-organization, in which a long-range spatial structuring emerges from short-range interactions and breaks the symmetry of the homogenous state. The mechanisms underlying self-organization range from self-assembly of equilibrium complexes to out-of-equilibrium formation of dissipative structures [88,89]. It is thus plausible that a variety of different mechanisms could be involved inside the cell.

## 7. Conclusions

The physical notion of phase separation opens novel research avenues in the field of transcriptional gene regulation by suggesting a possible interplay between assembly of nuclear bodies and recruitment of specific genomic sequences. However, it remains to be determined to what extent such interplay is dependent on phase separation, or on more complex active and/or specific processes. Here, HRS-seq, combined with other approaches, can be instrumental for dissecting the relationship between 3D chromatin organization and nuclear bodies, and its implication for both *cis*- and *trans*- co-regulation of gene expression. Understanding the relevance of phase separation in a biological context will require theoretical studies devising microscopic descriptions accounting for the intrinsic fluctuations present at the intracellular scale, as well as experimental studies investigating the possible involvement of active mechanisms.

## Figures and Tables

**Figure 1 genes-10-01049-f001:**
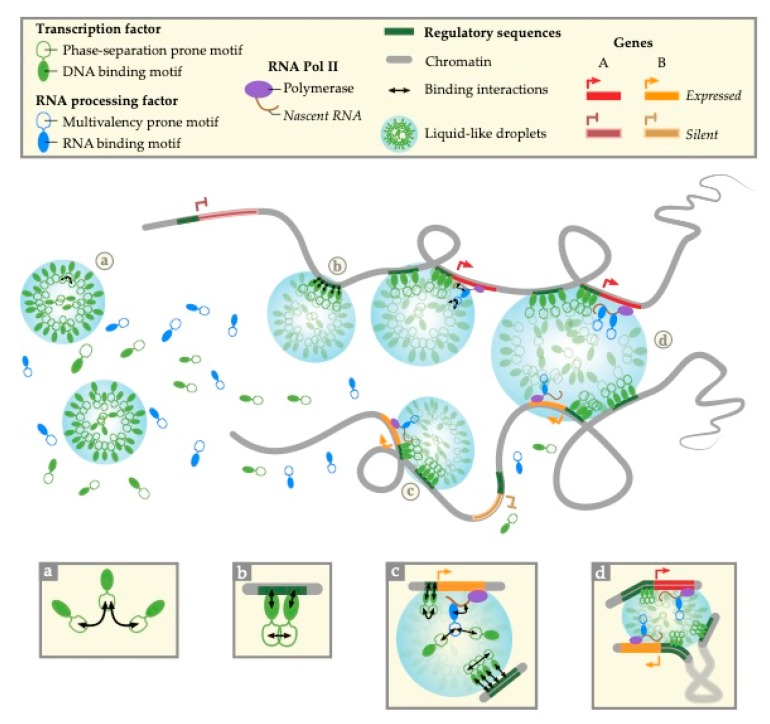
An integrated phase-separation model for self-assembly of transcriptional condensates controlling mammalian gene expression. (**a**) Transcription factors containing motifs prone to phase separation (e.g., IDR, Intrinsically Disordered Region) form liquid-like droplets (shaded in blue) by phase separation. (**b**) Their DNA binding motifs target specific genomic loci that are specifically incorporated into the droplets thus forming transcriptional condensates. Alternatively, phase separation could occur after binding of transcription factors (PPPS) on their target genomic sites in which case the corresponding DNA sequences act as nucleation sites. (**c**) Supplemented with the action of RNA processing factors containing motifs prone to multivalent interactions [67], they bring enhancers, promoters and/or nascent RNA transcripts in close vicinity, thus stabilizing long-range *cis*-interactions and promoting transcription. (**d**) In some instances, transcriptional condensates containing similar/compatible phase separation-prone motifs could finally merge into larger nuclear sub-organelles, leading to the formation of nuclear bodies like the Histone Locus Bodies (HLBs). The latter process brings together loci with similar transcriptional regulation but located on distinct (Topologically Associating Domains) TADs/chromosomes (orange/red lines and arrowheads), thus favouring the coordinated expression of the corresponding genes.

**Figure 2 genes-10-01049-f002:**
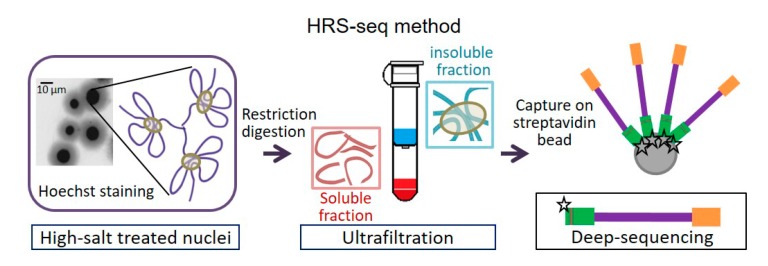
Principle of the HRS-seq method allowing the high-throughput identification of genomic sequences significantly associated with large RNP (ribonucleoprotein) complexes and nuclear bodies (adapted from [69]).

**Table 1 genes-10-01049-t001:** Classic nuclear bodies: main characteristics and components.

Compartment Name	Count/Nucleus	Diameter (µm)	Main Component	Main Associated Function	Ref.
Transcription factory	100	-	RNA Pol. II	mRNA transcription	[31]
Nucleolus	1–4	2–5	RNA Pol. I/Nucleolin	rRNA transcription	[32]
Cajal Body	10	0.5–1	Coilin, SMN ^1^	Splicing	[33]
Gem	10	0.5–1	SMN1	SMN sequestration	[34]
Histone Locus Body	2–4	0.5–1	Coilin, NPAT ^2^	Histone gene expression	[29]
Polycomb body	10–20	0.2–1.5	PRC1/PRC2 ^3^	Histone PTMs ^4^	[35]
PML body ^5^	10–20	0.1–1	PML	Apoptosis, viral defense	[36]
Nuclear speckle	20–50	2–3	SC35 ^6^, RNA Pol. II	Splicing	[37,38]
Paraspeckle	10–20	0.5–1	NEAT1 ^7^ lncRNA	Transcription	[39,40]

^1^ Survival of Motoneuron; ^2^ Nuclear protein, coactivator of histone transcription; ^3^ Polycomb repressive complexe; ^4^ PTM = Post-Translational Modifications; ^5^ Promyelocytic leukaemia nuclear body; ^6^ Serine/arginine-rich splicing factor; ^7^ Nuclear Paraspeckle Assembly Transcript 1.

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
