# Peer review of "Exploring Mammalian Genome within Phase-Separated Nuclear Bodies: Experimental Methods and Implications for Gene Expression"

_genes, 2019, doi:10.3390/genes10121049_

Round 1

Reviewer 1 Report

Lesne et al., Exploring mammalian genome within phase-separated nuclear bodies: implications for the regulation of gene expression

In this review manuscript, Lesne et al. described the recent progress in our understanding of 3D genome organization from the standpoint of nuclear bodies that are dynamically assembled and disassembled through phase separation mechanisms. They also speculated on the potential roles of nuclear bodies using data obtained from their recently established High-salt Recovered Sequences sequencing (HRS-seq) technology.

This article is comprehensive regarding the latest development in the field of phase separation mechanisms in chromosome biology and is overall well written, grammatically as well as scientifically. However, although I am not an expert of phase separation, I had three major concerns that I feel needs to be addressed.

On page 5 (lines 192–196), the authors came up with two predictions from the integrated phase-separation model of transcription condensates, with an intention to eventually understand the role of nuclear bodies in such processes. However, it was not clear to me what led the authors to bring up these two predictions. I may have misunderstood something, but it seems that there is a lack of explanation here, especially for those who are unfamiliar with phase separation. Please explain a bit more and clarify this point.

The effectiveness of HRS-seq in understanding problems associated with phase separation was also unclear. If I understand correctly, the HRS-seq method simply utilizes the classic nuclear halo assay, which has been around for years. Nuclear halo assay separates soluble and insoluble fractions of the nucleus by artificially treating nuclei with high salt (2M NaCl) and has been used to identify the so-called MARs (nuclear matrix attachment regions), which correspond to genomic regions that correspond to chromatin loop anchors enriched in the insoluble fraction. While a portion of the sequences recovered might actually correspond to those associated with ribonucleoprotein (RNP) complexes and nuclear bodies (Baudement et al., Genome Res 2018), it is by no means a method that is specifically targeted for these sequences and likely contains certain level of noise or non-specific binding. In that sense, I find the section (section 5) on HRS-seq to be rather misleading, giving a wrong impression that the HRS-seq is a method that is ideally suited to understand the roles of phase separation in chromosome biology. HRS-seq could be a good starting point, but it seems that one really needs to understand what exactly is being enriched by the high salt treatment in the insoluble fraction before comprehending the NGS data. I believe the authors should describe these aspects of HRS-seq and discuss not only the pros but also cons of HRS-seq with regards to its application to phase separation biology.

This is related to point 2, but the authors should also introduce other recent NGS-based technologies out there that can potentially be used to understand these levels of chromosome biology. Andrew Belmont’s TSA-seq (Chen et al., JCB, 2018) immediately comes to mind but is not discussed at all. TSA-seq seems more direct and informative, in my opinion, although it seems to require quite a lot of cells (the situation seems to be improved in the latest biorxiv preprint describing TSA-seq 2.0). Maybe there are other methods in the literature that I am not aware of that are worth mentioning in this article.

Other points:

p.1 Abstract, line 22–23: “High-salt Recovered Sequences sequencing (HRS-seq), designed to identify chromatin regions associated with large ribonucleoprotein (RNP) complexes and nuclear bodies” > HRS-seq relies on a method that aimed to identify MARs but it happened that the recovered sequences contained RNPs and nuclear bodies (Engelke et al., J Proteome Res 2014; by the way, this article should be referenced). I don’t believe it was originally “designed” to identify such sequences so the sentence should be rephrased.

p.1 Abstract, line 25–26: “The development of novel experimental approaches, including our HRS-seq method” > I did not see approaches other than HRS-seq being included in the main text. As I pointed out, how about TSA-seq? The author should provide an unbiased view of the latest development in the field, not just their own.

p.2, line 45: developed : > developed: (erase the space in between)

p.2, line 58–61: “While these droplets display…” > sentence too long and difficult to understand. The sentence needs to be rephrased

p.2, line 73: “(3C/Hi-C), has revealed that, beyond the…” > commas unnecessary

p.2, line 78: “old distinction” > “classic distinction”

p.2, line 81: Along with references 13 and 14, Rao SSP et al., Cell 2017 “Cohesin loss eliminates all loop domains” should be referenced.

p.2, line 85: Along with or instead of reference 16, Dixon et al., Nature 2012 and 2015 from Bing Ren’s laboratory should be cited here. They were the first to show this.

p.2, line 89: At the end of this paragraph, the authors could add a discussion of papers that have addressed the initial formation of TADs and compartments during early embryogenesis, mainly in mice (Tachibana lab, Wei Xie lab etc.). Same can be done for papers like the followings that are starting to address some aspects of compartment regulation principles: Bonev et al, Cell 2017; Stadhouders et al., Nat Genet 2018, Miura et al., Nat Genet 2019.

p.6, line 228: nuclease treatments > it was puzzling how one can combine nuclease treatment with an NGS-based technology like HSR-seq. Please explain.

p.7, line 253: “this” > should be deleted

Reviewer 2 Report

The review by Lasne and colleagues explores the possible link between the assembly of phase separated nuclear condensates and long-range genomic interactions. It proposes that the combination of new methods based on HRS-seq will help understand whether phase separation influences genome organization at the supranucleosomal scale, particularly TADs and chromosomal compartments.

Major comments:

The authors should comment on the current debate about the liquid-like nature of transcriptional condensates and more clearly explain how evidences indicating that such bodies are phase separated in cells are still limited. The authors use caution by defining transcriptional bodies as “liquid-like phase separation” rather than LLPS. Yet, the term used by the authors is misleading (as it suggests a liquid nature) and it would be more correct to use the term “hub” when referring to in cells mechanisms throughout the manuscript. Similar corrections should be made for the term “transcriptional condensates” (Banani et al. 2017; McSwiggen et al. 2019 doi: 10.1101/gad.331520.119).

The authors should clarify throughout the text which statements regarding phase separating molecules are based on experimental data and which statements are speculations. In line with this, the authors should carefully make distinctions between in vitro and in vivo observations when they report published work. This is important because in vitro condensates may not be bona fide condensates in vivo. For example, the roundness and fusion of puncta in cells are not always indication of phase separation (McSwiggen et al. 2019 doi: 10.7554/eLife.47098). It would help if the authors clearly describe the state-of-art in the field and make a distinction between well characterized phase separated condensates in cells (for example P granules), and others that have been mainly only characterized in vitro with weaker evidence in vivo (such as Mediators and transcriptional hubs).

Minor comments:

From lines 124-128, it is not clear whether HP1 phase separation has been shown in vivo. It has been shown that HP1 forms liquid droplets both in vitro and in vivo (Larson et al. 2017; Strom et al. 2017). In particular, Strom et al. performed many in vivo experiments that who the in vivo liquid properties of HP1 puncta (see criteria discussed in McSwiggen et al. 2019 doi: 10.1101/gad.331520.119). Further, with “it remains unclear whether, in vivo, heterochromatin domains actually rely on liquid-liquid phase separation”, it is not clear whether the authors refer to the fact that (i) it is not clear if HP1 forms condensates in vivo, or (ii) it is not clear if gene repression requires phase separation of HP1. These statements should be clarified in the text.

In lines 128-138, the authors make a distinction between LLPS and PPPS. This distinction might be too reductive as compare to the complexity of biological macromolecular condensates in general, and specifically in the context of chromatin. For example, it is possible that both LLPS and PPPS could be involved in the formation of condensates. This indeed is likely the case in the context of HP1 - it is known that: (i) HP1 proteins can bridge nucleosomes (Canzio et al. 2013; Canzio et al. 2011; Machida et al. 2018), but also (ii) HP1 self-oligomerization driven by phosphorylation is sufficient to drive HP1 phase separation. Further HP1 phases, can incorporate chromatin (Larson et al 2017). See more details in Peng and Weber 2019 https://doi.org/10.3390/ncrna5040050. Because this section in the manuscript might give rise to confusions and misleading conclusions, it should be justified by the authors why they choose such a reductive approach or more possibilities should be discussed.

Another aspect that is not taken into account by the authors is that chromatin can phase separate on its own and contribute to multi-valency (Gibson et al, 2019; Sanulli et al, 2019). This aspect adds an extra layer of complexity into the system and authors should comment on how this integrates in their model.

The statement in lines 147-150 is confusing for two aspects: Transcriptional condensates have been proven to phase separate in vitro, but in vivo data are still preliminary because the small size of the condensates prevents accurate analysis (as pointed out by the authors line 157-160). From the statement, it seems that it has been proven that heterochromatin phases (HP1) are PPPS, which is not the case (Peng and Weber, 2019 https://doi.org/10.3390/ncrna5040050).

The integrated model described in line 192-196 should include the possibility of both PPPS and LLPS occur within a condensate.

As discussed by the authors in lines 229-232, 1,6 hexanediol disrupts hydrophobic interactions. However, LLPS can also be driven by electrostatic interactions (as also mentioned by the authors in line 161-162) and electrostatic interactions are not disrupted by 1,6 hexanediol – rather by salt, which would make them hard to be detected by HRS-seq in 2M salt (see details in Alberti et al. 2019 doi: 10.1016/j.cell.2018.12.035). Therefore, 1,6 hexanediol alone is not a good tool to discriminate LLPS and PPPS. Further, HRS-methods may not be able to detect electrostatic driven phase separated condensates.

Lines 202-204, “We have recently shown that performing a liquid (or liquid-like) to solid phase transition through high-salt treatments of transcriptionally active nuclei makes large RNP complexes, including nuclear bodies, insoluble.” This sentence is misleading as the physical properties of the materials – before and after precipitation - have not been addressed in Baudement et al. 2018. (Note that precipitation is not synonymous of solid – liquid droplets (denser phase) also sediment). I would rephrase as: “We have recently shown that performing high-salt treatments of transcriptionally active nuclei makes large RNP complexes, including nuclear bodies, insoluble.”

In lines 209-210, the authors should clarify that HRS-seq obtains a global profiling of all RNP-associated sequences, not only nuclear bodies (as reported in Baudement et al. 2018). And that the method cannot discriminate between phase separated condensates or a hub of proteins.

Reviewer 3 Report

Lesne et al present a review on liquid-like phase transitions (LLPS) and polymer-polymer phase separation in nuclear organization. In particular, they highlight a sequencing method that they have developed to extract sequences that are associated with such sub-nuclear organelles.

This review is valuable to the community as it presents an overview of mechanisms that are likely to drive LLPS and PPPS phenomena and methods with which to distinguish them.

There are some minor comments that the authors should address:

The authors should comment on the limitations and possible extensions of the HSR method. For example it seems that HRS surveys multiple types of subnuclear compartment in the same experiment – are there likely to be any modifications that would allow segregation of the data by compartment? Is the method applicable to following dynamic changes in the sub-compartments that would inform on gene expression? Line 228, the authors note that distinguishing PPPS from LLPS can be done by using a nuclease to remove DNA. However, as the authors point out, many of these compartments are organized around RNPs. In this case presumably removing RNA is also important and should be noted here. Also, it would be helpful to note at which point in the protocol the nuclease is added. Is this before salt precipitation or afterwards and why? Line 231, Although IDRs are required for formation of LLPS, many of these proteins also have folded domains. Is it known whether these remain folded on hexanediol treatment? Can the authors comment on how this might affect the outcome/interpretation of such experiments? Line 427, the authors note that the formations of phase separated compartments depends on the “local” concentration of the components. Please clarify what is meant by “local” in this context. It is likely that during PPPS  high local concentrations are induced on the polymer through specific interactions. However, for LLPS, it seems more likely that concentration at the level of a whole cellular compartment, such as the nucleus is more relevant. Line 271, the authors note that current thermodynamic models of LLPS formation are useful but may not provide a full description because in some cases the numbers of molecules involved are small and so stochastic effects will impact on the likelihood of formation of phase separated domains. It would be helpful if they could elaborate on how this stochasticity is likely to impact on model predictions and/or measurement of phase separation phenomena.

Reviewer 4 Report

This review Lesne et al. is very well-written and is a smooth read. The content is organized. The review is concise, which is both a strength and a weakness. It seems that the primary motivation of the authors to write the current review is to highlight their HRS-seq method, published in 2018 in Genome Research. It is an interesting method, so this is, in principle, a good idea, although the review could be deeper with respect to describing their method in ways that go beyond their Genome Research article. And the title of the review is misleading in that the review title can be viewed as at least a bit incongruous with what is actually the main focus of the work.

Although it is somewhat out of my field, I did not detect obvious misrepresentation of prior work, although I was surprised to not see citations to papers such as Kato et al., McKnight Cell 2012 or Jin et al., J.K. Kim, Cell Reports 2017.

There are some aspects of Figure 2 that seems identical to parts of Figure 1 of the authors' Genome Research method article; I am not sure if this is a copyright issue with that CSHL Press journal, but it may warrant at least some  changes to the current Figure 2.

Round 2

Reviewer 2 Report

Overall the manuscript is much improved. However, the authors still use with imprecision and confusion the terms phase separation/hub/condensate.

Condensates are defined in the field as phase separated assemblies.

Hubs are normally defined as groups of proteins that cluster together through unknown mechanisms. Importantly hubs might not be driven by phase separation.

The authors introduce the term “phase separated hubs”. This is confusing and contradictory. If the authors describe a phase separated system, the term condensate should be used. If the authors describe a cluster of proteins of unknown entity, they should not use the term “phase separated” but only hub or (nuclear) body. If the authors want to use these words with a different meaning, they should clarify it.

Other comments:

Table 1: enumerates classic nuclear bodies. Stress granules should not be in the list as they are found in the cytosol.

Line 135-140: LLPS and PPPS should be defined properly before describing it in the context of HP1 (the authors do that later in the text). This is necessary because the distinction between LLPS and PPPS is not widely used in the field.

Similarly, the term “collapse” used in line 141 should be defined. Indeed, it can have different meaning for physicists and chromatin biologists.

Line 156 The word condensate should be used not hub, since these in vitro data show the phase separated nature of the material.

Line 157 “that transcription factors activate genes through the phase-separation capacity of their activation domains”. I believe these refers to in vivo studies not in vitro (not clear from the sentence). Even if this sentence is the title of the cited manuscript, the authors should be careful in the statement and rather say something like “domains driving phase separation in vitro are also important for gene activation in vivo”.

Line 159-163 “Therefore, it is not yet clear whether, in vivo, phase separation truly relies here on a liquid-like phase separation similar to the LLPS observed for larger nuclear compartments like the nucleolus, or whether it reflects a hybrid situation also involving a polymer collapse process and PPPS as suspected in the case of heterochromatin domains.”

I believe that the most important point here is not whether transcriptional condensates are formed by LLPS or PPPS, but more in general if they are phase separated in vivo. This sentence can be misleading as it seems to assume that transcriptional bodies are phase separated. I would replace with something like “It is not clear whether the formation of transcriptional hubs in vivo relies on phase separation processes, either LLPS or PPPS.”

Line 166-168. The authors should better explain why this observation in the context of HSV1 infection is relevant in the context of transcriptional bodies. This work on HSV1 shows that the formation of clusters of PolII in cells can be explained by mechanisms that are different from phase separation. The same types of experiments have not been done in the context of PolII in absence of viral infection (and likely they cannot be done because of the smaller size of the bodies). Hence, the debate about transcriptional bodies being bona fide phase-separated condensates in vivo.

Line 173-175. This distinction seems vague and not accurate (bridging factors can also be multivalent and weak binders). Defining what the authors intend with LLPS and PPPS as suggested above can help clarify.

Line 190-194 and Line 223-225 contain contradictory terms condensate/hub/phase separation as explained above.

Figure 1 legend: Contradictory terms are used for condesate/hub. If described as a speculative model and if they intend to refer to only phase separated compartments, the authors should use the term condensates. Otherwise, both hubs and condensates could be part of the model. I believe the model could apply to any type of nuclear body, either formed by phase separation or by other mechanisms.

Line 292-294 “LLPS and liquid-like phase separation should be sensitive to compounds that disturb hydrophobic interactions, like 1,6 hexanediol, unlike PPPS”.

This statement is not correct: PPPS driven by hydrophobic interactions will also be disrupted. 1,6 hexanediol can identify the genomic content of nuclear bodies whose formation is driven by hydrophobic interactions (not only phase separated). All other types of interactions will not be directly affected by the chemical.

Line 316 Phase separation depends on the local concentration – it was correct in the previous version.

Line 370 As explained above, I suggest replacing “RNP phase separation” with RNP nuclear bodies.

Response to the authors comment "The physical concept of “phase separation” is useful when it involves a simple mechanism, basically when the affinity between the molecules of the considered species is higher than the affinity between these molecules and their surroundings (solvent). When more complex mechanisms are involved (like chromatin compaction discussed in Gibson et al., 2019), it is more relevant to use the concepts of self-assembly (towards an equilibrium state) or self-organization (towards an out-of-equilibrium state)."

I agree that phase separation occurs when the affinity between the molecules of the considered species is higher than the affinity between these molecules and their surrounding. I disagree that phase separation does not apply to chromatin. Gibson et al.and Sanulli et al. show that nucleosomes within chromatin fibers mediate the multivalency that drives phase separation. In other words, nucleosomes preferentially interact with nucleosomes rather than with the solvent. I believe that authors refer to the fact that, in the chromatin compaction context, nucleosomes are in some case part of the same fiber as compare to “isolate” molecules such as HP1. Yet, chromatin alone can form phase separated condensates in vitro, very much like proteins such as HP1 or PolII. It is also remarkable that “isolate” molecules could also for oligomers (for example HP1) and therefore become similar to connected nucleosomes in a chromatin fiber. This is a minor point and might not be included in the manuscript if the authors believe is beyond the scope of their work.

Author Response

REVIEWER 2 (ROUND 2) (reviewer’s comments are in italics, author responses are in bold);

Overall the manuscript is much improved. However, the authors still use with imprecision and confusion the terms phase separation/hub/condensate.

Condensates are defined in the field as phase separated assemblies. Hubs are normally defined as groups of proteins that cluster together through unknown mechanisms. Importantly hubs might not be driven by phase separation.

We carefully checked our manuscript for the terms “hub” and “condensate” and made the recommended corrections in order to use the right term in the right context. We kept the term “hub” in lines 64, 104, 159, 165, 166, 172, 174, 190, 193, 280, where the involvement of phase separation was not clear, and changed it for “condensates” elsewhere. This terminology is now explained in lines 63-64.

The authors introduce the term “phase separated hubs”. This is confusing and contradictory. If the authors describe a phase separated system, the term condensate should be used. If the authors describe a cluster of proteins of unknown entity, they should not use the term “phase separated” but only hub or (nuclear) body. If the authors want to use these words with a different meaning, they should clarify it.

As stated above, we have now adopted the terminology recommended by the reviewer and paid better attention to use the appropriate term in the right context. In particular, we have corrected the misleading formulation “phase-separated hubs” and replaced it by “phase-separated condensates” (Line 295).

Other comments:

Table 1: enumerates classic nuclear bodies. Stress granules should not be in the list as they are found in the cytosol.

We are grateful to the reviewer for pinpointing this mistake. Stress granules have been removed from Table 1. 

Line 135-140: LLPS and PPPS should be defined properly before describing it in the context of HP1 (the authors do that later in the text). This is necessary because the distinction between LLPS and PPPS is not widely used in the field.

We thank the reviewer for pointing this lack of consistency in the presentation. As requested, LLPS and PPPS definitions have been moved before the description of the HP1 context (Lines 132-139).

Similarly, the term “collapse” used in line 141 should be defined. Indeed, it can have different meaning for physicists and chromatin biologists.

The term ‘collapse’ indeed deserved an explanation as it was here used with its meaning in polymer physics. The collapse of a polymer (here the chromatin fiber) is simply a change of its shape (a ‘conformational transition’) leading to an increase of its local density. It can be due to bridging molecules binding the fiber or to hydrophobic interactions between the monomers (or both). In any case the term ‘collapse’ reflects the interplay between local interactions and the fiber connectedness. Explanations have been added (Lines 133-136).

Line 156 The word condensate should be used not hub, since these in vitro data show the phase separated nature of the material.

 We changed this word as requested (Line 158).

Line 157 “that transcription factors activate genes through the phase-separation capacity of their activation domains”. I believe these refers to in vivo studies not in vitro (not clear from the sentence). Even if this sentence is the title of the cited manuscript, the authors should be careful in the statement and rather say something like “domains driving phase separation in vitro are also important for gene activation in vivo”.

This sentence has been changed as recommended (Lines 158-159).

Line 159-163 “Therefore, it is not yet clear whether, in vivo, phase separation truly relies here on a liquid-like phase separation similar to the LLPS observed for larger nuclear compartments like the nucleolus, or whether it reflects a hybrid situation also involving a polymer collapse process and PPPS as suspected in the case of heterochromatin domains.”

I believe that the most important point here is not whether transcriptional condensates are formed by LLPS or PPPS, but more in general if they are phase separated in vivo. This sentence can be misleading as it seems to assume that transcriptional bodies are phase separated. I would replace with something like “It is not clear whether the formation of transcriptional hubs in vivo relies on phase separation processes, either LLPS or PPPS.”

We agree with the reviewer. This sentence has been modified as recommended (Lines 160-164).

Line 166-168. The authors should better explain why this observation in the context of HSV1 infection is relevant in the context of transcriptional bodies. This work on HSV1 shows that the formation of clusters of PolII in cells can be explained by mechanisms that are different from phase separation. The same types of experiments have not been done in the context of PolII in absence of viral infection (and likely they cannot be done because of the smaller size of the bodies). Hence, the debate about transcriptional bodies being bona fide phase-separated condensates in vivo.

This work is mentioned here as an example to illustrate the fact that some clusters/hubs of RNA polII do form by mechanisms different from phase separation. This example is intended to be a caveat, prompting to be careful before considering a transcriptional hub as a condensate formed by phase separation. We added a sentence (Lines 164-165) to clarify that our scope is here transcriptional hubs in general: “In all instances, we should remain careful before considering a transcriptional hub as a condensate formed by phase separation”. 

Line 173-175. This distinction seems vague and not accurate (bridging factors can also be multivalent and weak binders). Defining what the authors intend with LLPS and PPPS as suggested above can help clarify.

As defined above and now in the text (Lines 132-139), while LLPS occurs through demixing of two liquid/liquid-like phases under the effect of weak multivalent interactions, while PPPS is thought to involve the binding of bridging factors on a polymer, here the chromatin fiber. Considering the connectedness of the chromatin fiber, PPPS is better described as a polymer collapse. The distinction between LLPS and PPPS has been clarified in the revised version, based on the discussion in ref.54.

Line 190-194 and Line 223-225 contain contradictory terms condensate/hub/phase separation as explained above.

 As previously mentioned, we modified these sentences as requested (Lines 193-197 and 224-231).

Figure 1 legend: Contradictory terms are used for condensate/hub. If described as a speculative model and if they intend to refer to only phase separated compartments, the authors should use the term condensates. Otherwise, both hubs and condensates could be part of the model. I believe the model could apply to any type of nuclear body, either formed by phase separation or by other mechanisms.

As stated in line 198, we propose this model as a working hypothesis involving phase-separated compartments. Therefore, as suggested, we corrected the caption of figure 1 legend to use “condensate” (Lines 209, 213, 219), as this is here more appropriate than “hub”.

Line 292-294 “LLPS and liquid-like phase separation should be sensitive to compounds that disturb hydrophobic interactions, like 1,6 hexanediol, unlike PPPS”. This statement is not correct: PPPS driven by hydrophobic interactions will also be disrupted. 1,6 hexanediol can identify the genomic content of nuclear bodies whose formation is driven by hydrophobic interactions (not only phase separated). All other types of interactions will not be directly affected by the chemical.

We agree that in vitro the 1,6 hexanediol, when used at high concentrations, would probably disrupt even the strong hydrophobic interactions possibly involved in PPPS in vivo. However, under the experimental conditions used in vivo, only the weakest hydrophobic interactions, like those inducing LLPS, are expected to be affected (see ref.80). As previously explained, PPPS is thought to involve only strong interactions (electrostatic or hydrophobic) that are not expected to be affected by the smooth experimental conditions used in vivo. To clarify this point, we changed this sentence as follows (Lines 288-291): “Therefore, LLPS and liquid-like phase separation should be sensitive to compounds that disturb weak hydrophobic interactions, like moderate 1,6 hexanediol treatments [80], unlike PPPS that relies on stronger interactions”.

Line 316 Phase separation depends on the local concentration – it was correct in the previous version.

The word “local” was also unclear to another reviewer, hence our change. We now formulate the sentence as: “Phase separation depends on the local concentration within the nucleus (or a region of the nucleus) …” (Lines 307-309).

Line 370 As explained above, I suggest replacing “RNP phase separation” with RNP nuclear bodies.

 We agree with the reviewer and changed “RNP phase separation” by “nuclear bodies” (Line 357).

Response to the authors comment "The physical concept of “phase separation” is useful when it involves a simple mechanism, basically when the affinity between the molecules of the considered species is higher than the affinity between these molecules and their surroundings (solvent). When more complex mechanisms are involved (like chromatin compaction discussed in Gibson et al., 2019), it is more relevant to use the concepts of self-assembly (towards an equilibrium state) or self-organization (towards an out-of-equilibrium state)."

I agree that phase separation occurs when the affinity between the molecules of the considered species is higher than the affinity between these molecules and their surrounding. I disagree that phase separation does not apply to chromatin. Gibson et al. and Sanulli et al. show that nucleosomes within chromatin fibers mediate the multivalency that drives phase separation. In other words, nucleosomes preferentially interact with nucleosomes rather than with the solvent. I believe that authors refer to the fact that, in the chromatin compaction context, nucleosomes are in some case part of the same fiber as compare to “isolate” molecules such as HP1. Yet, chromatin alone can form phase separated condensates in vitro, very much like proteins such as HP1 or PolII. It is also remarkable that “isolate” molecules could also for oligomers (for example HP1) and therefore become similar to connected nucleosomes in a chromatin fiber. This is a minor point and might not be included in the manuscript if the authors believe is beyond the scope of their work.

We thank the reviewer for sharing this insightful view. What we meant in our response was merely that in the case of chromatin fiber, the interplay between the fiber connectedness and the local interactions between the nucleosomes and the solvent leads to a far more complex situation. However, a discussion of this point is indeed beyond the scope of our present work.

We thanks this reviewer for his/her useful criticism and suggestions that greatly improved our manuscript.

Reviewer 4 Report

The authors have added some sentences/paragraphs to go deeper into the impact of their new method and how it relates to other methods in the field. They also have made other small changes throughout to address the reviewers, including my critiques. Overall, the added text improves the manuscript and makes it clearer that the focus of this work is largely on methods in the field.

My one remaining concern is that the authors did not address my main concern from the initial review: i.e., the title. It seems to me that the title should include the words "methods" or "techniques"  or something like this to convey to potential readership that this review is largely focused on experimental approaches/methods. The experimental approaches are the authors' main rationale for, and focus of, this review, so it seems to me that the title needs to convey this. This could be accomplished by replacing "Exploring" in the title with "Experimental approaches for studying…" or something to this effect.

Author Response

REVIEWER 4 (ROUND 2) (reviewer’s comments are in italics, author responses are in bold);

Comments and Suggestions for Authors

The authors have added some sentences/paragraphs to go deeper into the impact of their new method and how it relates to other methods in the field. They also have made other small changes throughout to address the reviewers, including my critiques. Overall, the added text improves the manuscript and makes it clearer that the focus of this work is largely on methods in the field.

My one remaining concern is that the authors did not address my main concern from the initial review: i.e., the title. It seems to me that the title should include the words "methods" or "techniques" or something like this to convey to potential readership that this review is largely focused on experimental approaches/methods. The experimental approaches are the authors' main rationale for, and focus of, this review, so it seems to me that the title needs to convey this. This could be accomplished by replacing "Exploring" in the title with "Experimental approaches for studying…" or something to this effect.

We apology for missing this point in our previous answer. Following reviewer recommendation, we added the words “experimental methods” to our title: “Exploring mammalian genome within phase-separated nuclear bodies: experimental methods and implications for gene expression”.